



# System identification of offshore wind turbines for model updating and validation using field measurements

Jakob Gebel[1,2], Ashkan Rezaei[1], Adithya Vemuri[2], Veronica Liverud Krathe[1], Pieter-Jan Daems[2], Jens Jo Matthys[2], Jonathan Sterckx[2], Konstantinos Vratsinis[2], Kayacan Kestel[2], Amir R. Nejad[1], and Jan Helsen[2]

[1]Department of Marine Technology - Norwegian University of Science and Technology (NTNU), Trondheim, 7052 Trøndelag, Norway
[2]Department of Mechanical Engineering - Vrije Universiteit Brussel (VUB), Brussel, 1050 Elsene, Belgium

**Correspondence:** Jakob Gebel (Jakob.Gebel@ntnu.no)

**Abstract.**

This study presents an applied system identification approach for developing, updating, and validating simulation models of wind turbines using field measurements. This is demonstrated by developing a model of a bottom-fixed offshore turbine in the Belgian North Sea. An initial model is obtained based on available design information and the scaling of reference models. Afterwards, the model is calibrated by leveraging drivetrain vibration data and blade strain measurements. This is accomplished using Operational Modal Analysis techniques, which enable the identification of the turbine's modal parameters, particularly its eigenfrequencies. By identifying the eigenmodes, the model can be updated to match the modal behaviour of the deployed turbine. Comparison with SCADA data further validates the model's operational performance. Additionally, the obtained model is used to calculate blade pitch-bearing lifetime estimations in different wind conditions, to give insight in the detrimental effect of wind properties on the pitch-bearing lifetime, which can be used as decision support in operation and maintenance strategies for wind turbines.

## 1 Introduction

Within the European Green Deal, offshore wind is seen as a core energy source by the European Commission and is set to amount for 111 GW across all EU sea basins by 2030 (European Commission, 2023). This requires installed offshore wind capacity in the EU to increase sevenfold within the next six years. Similar ambitious goals are set in the USA, China and Japan, according to DOE (2023), Yang et al. (2016), and Mogi (2022) respectively. To facilitate this growth, it is necessary to further reduce the levelized cost of electricity (LCOE) for offshore wind turbines. Currently, the LCOE of offshore wind remains more than double the costs for onshore wind, as outlined by the International Renewable Energy Agency (2023). An important factor are the high operation and maintenance (O&M) costs of offshore wind farms, which can constitute up to 30% of the LCOE of offshore wind turbines (Röckmann et al., 2017).

While physics-based simulation models are a well-established tool used by turbine manufacturers to reduce the costs of design and certification processes, they also emerge as a promising tool to lower O&M costs by providing additional decision




support. First off, energy production estimates can be obtained by simulating the aerodynamic behaviour of multiple wind turbines simultaneously, accounting for potential interactions among them, such as wake loss effects. This is used to optimise

the farm layout in the design phase and for farm control in the operational phase. Wake steering, axial induction control and general yield studies can maximise the energy production of an already installed wind farm (van Binsbergen, 2020). Additionally, dependable forecasts of both short and long-term energy production capabilities can be obtained, which is essential for operators participating in bidding processes. An overview of farm layout optimization is provided by Hou et al. (2019), while Qian and Ishihara (2021) provide a wake-steering optimization to maximize wind farm power production.

Second, structural behavior is analysed by modelling individual turbines to assess their response under various environmental conditions. This enables operators to pinpoint particularly detrimental conditions for their assets and formulate strategies to mitigate structural damage. Smilden et al. (2017) give an example by identifying the key contributors to lifetime accumulated fatigue damage for the support structure of a 10 MW offshore wind turbine.

Lastly, the condition of individual turbine parts can be assessed through simulations to estimate their remaining useful life

(RUL). This information is utilized to guide maintenance strategies and enables the quantification of RUL for potential lifetime extensions at the end of a turbine's service life or identifying components prone to failure within a system. Providing a RUL estimate for various turbine components can also be used to guide maintenance schedules and lower the down times associated with O&M. Nejad et al. (2014b) demonstrate how load series obtained from a multibody simulation can enable a reliability-based maintenance plan for a wind turbine gearbox by identifying components with the highest probability of failure.

Additional information about the use of physics-based simulation models for wind turbines can be found in Otter et al. (2022) and Greco et al. (2023). Alternative ways to obtain decision support include data-driven techniques, which apply machine learning algorithms and statistical models on large historical datasets to extract valuable insight. For a comprehensive review of data-driven approaches, readers are referred to Pandit et al. (2023). Unlike data-driven techniques, physics-based simulations do not require a large amount of historical data for model training and allow prediction of the turbine's behaviour across

various environmental conditions, even in instances where such conditions have not been previously observed.

Leveraging the advantages of physics-based simulation models presents a challenge for farm operators and researchers, as they seldom have access to existing simulation models or all required design information to build such models. Thus this paper showcases how a physics-based simulation model of a deployed wind turbine can be developed using a combination of scaling methods and field measurements. An aero-hydro-servo-elastic simulation model is obtained based on available design

information and scaling of reference turbines. The modal behaviour of the deployed turbine is identified through operational modal analysis (OMA) applied to strain and vibration measurements, enabling tuning of the model to ensure it represents real-world behaviour. Afterwards, available data of the Supervisory Control and Data Acquisition (SCADA) system is used to validate the operational performance.

OMA is a method used to identify the modal parameters of a structure from measurement data collected while the structure is

in its operational state, without relying on known external excitation sources. The identified modal parameters primarily consist of the natural frequencies, damping ratios, and mode shapes of the structure and describe the system's dynamic behaviour (Brincker and Ventura, 2015). Generally, identifying and monitoring the modal parameters of wind turbines can be used for





condition monitoring purposes, as changes in modal behaviour can be indicative for degradation processes. An example of this application is shown by Cadoret (2023), outlining the challenges of OMA application for wind turbines and identifying

damaged turbine blades based on changes in modal parameters. More important for the presented work, identifying modal parameters also allows for assessing a simulation model's validity. Page et al. (2019) demonstrate this approach by evaluating different foundation modelling approaches based on identified eigenfrequencies of the support structure. The presented work applies a similar approach, extending it to the complete turbine to identify the eigenfrequencies of the tower and rotor and to calibrate a simulation model. For this, an OMA method in frequency domain is used to identify eigenfrequencies of a wind

turbine based on the spectral density of the strain and vibration measurements. The identified values are used to tune and validate the simulation model's modal behaviour against the deployed turbine. Finally, the previously mentioned SCADA data is used to obtain power, pitch, and RPM curves of the deployed turbine, which are compared to the operational behaviour of the obtained model. The finished simulation model then allows to calculate damage estimates for turbine components and link the components' degradation to inflow conditions to identify and quantify the detrimental effect of different environmental

conditions and operational states. To showcase this, the last part of this paper describes how damage prognostics are obtained for the blade bearing. Here, load time series from the simulation are used with an analytical damage model, to provide blade bearing damage prognosis, i.e RUL estimates.

The main contribution of this paper lies in showcasing how field measurements can be used to obtain, tune, or validate simulation models of deployed turbines and showcase how to link the degradation of components to environmental conditions

and operational states. The pitch-bearing calculation offers a case study for applying a physical model to obtain insights into the detrimental effects of various wind properties. This approach enhances the understanding of pitch system failures, which are among the largest contributors to turbine downtime (Santelo et al., 2022).

The paper is structured as follows: Sect. 2 details the development of the simulation model. First, showing the applied scaling methods for the construction of an initial model. Then explaining the applied OMA methods and tuning the dynamic behaviour

of the model to the obtained results. The last part of Sect. 2 focuses on the validation of the operational behaviour of the model. In Sect. 3 the damage prognostic for blade pitch bearings is described and the influence of different turbulent classes and wind shear profiles on the expected pitch bearing lifetime is analysed.

## 2 Turbine model

The following sections describe how the simulation model of a deployed offshore wind turbine is obtained. The target turbine

has a power capacity of around 8 MW and is installed on a monopile in the Belgian North Sea. An initial model of the turbine is built in the multibody simulation software SIMPACK and then transferred to the aero-hydro-servo-elastic-solver OpenFAST. Modelling in SIMPACK first, allows for direct visualization of the model and its eigenmodes and is not dependent on mode shape coefficients as an input. The model is transferred to OpenFAST as the high computational costs of SIMPACK do not allow for simulations in real-time, which is essential for the application in digital twin frameworks in future work.

Figure 1 provides an overview of the modelling and validation process. The initial turbine model is obtained by scaling open-





source designs, based on the available design data of the target turbine (Sect. 2.1). A range of systems identification techniques are used to identify natural frequencies of the real turbine (Sect. 2.2). Herein eingenfrequencies of tower and blades are obtained from vibration data and blade strain measurements. The eigenfrequencies of the initial model are compared to the identified properties of the real turbine and the model is adjusted in multiple steps to match the observed behaviour (Sect. 2.3). Finally,

representative operational behaviour of the model is ensured by comparing simulation results to SCADA and vibration data (Sect. 2.4).

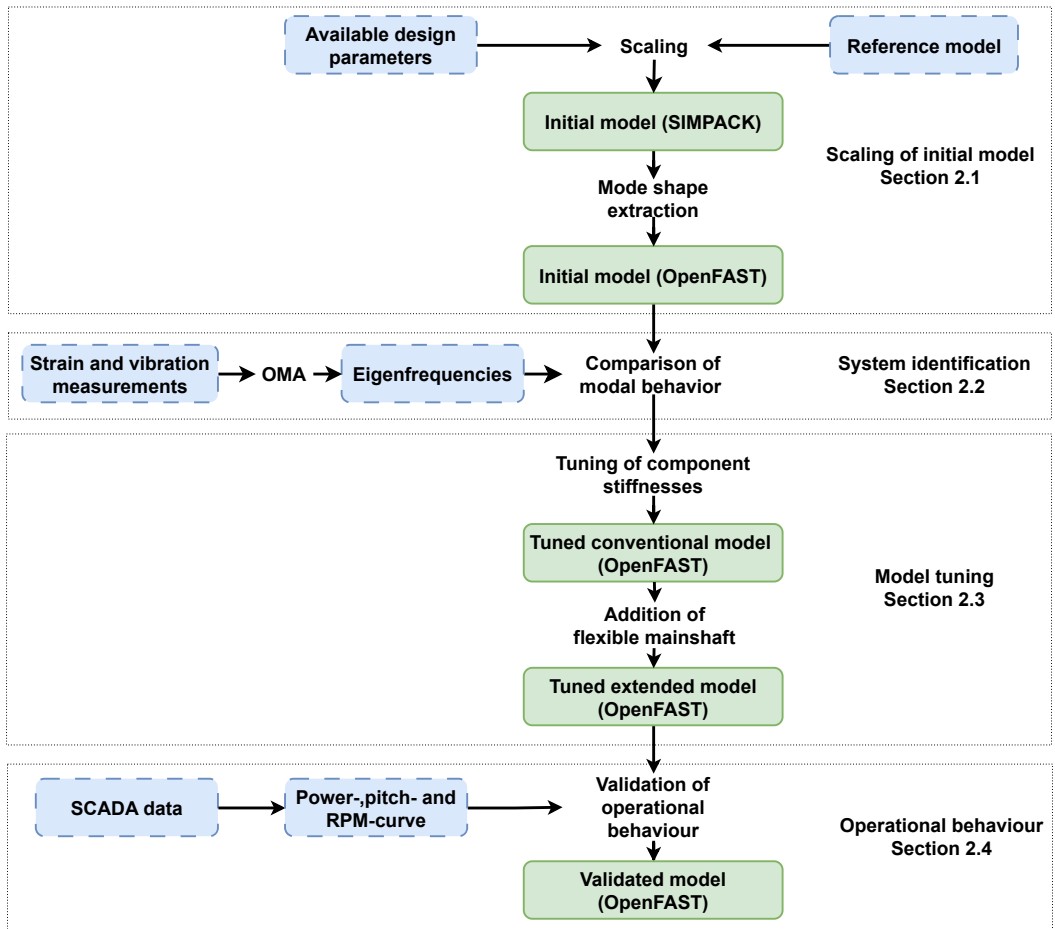

**Figure 1.** Schematic overview of modelling process annotating the corresponding sections. Model stages are depicted in green (solid) and utilised information and data in blue (dashed).





## 2.1 Scaling of initial model

The following subsections describe the scaling approaches used for each component of the turbine model. An overview of the resulting model is given in Figure 2, indicating the source of origin for the initial design of each component.

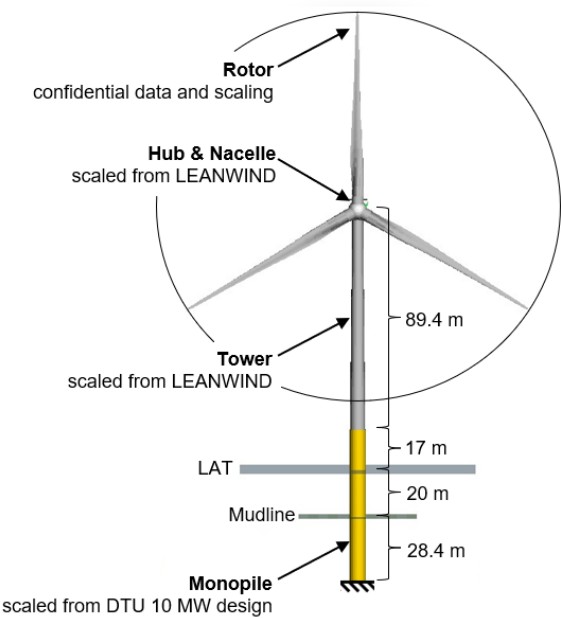

**Figure 2.** Overview of the obtained turbine model, indicating the source of origin for each component

### 2.1.1 Tower design

The available design information of the tower of the deployed turbine are the yaw bearing height and the height at the transition piece as well as the respective tower diameters at these points. To obtain an initial design based on this information, the tower of the LEANWIND 8 MW reference turbine is used, as described by Desmond et al. (2013). The LEANWIND turbine is a validated open-source design and is chosen as the reference model due to its similar power range and design. A bottom section of the LEANWIND tower is removed, retaining the upper 86.36 m to match the tower length of the target design. The top and bottom diameters are adjusted to match the given design specifications. The tower bending stiffness (flexural rigidity) is scaled based on rated thrust and new wall thicknesses are calculated accordingly, assuming the same material as in the LEANWIND design. The tower coning is assumed to be linear, as seen in the reference models, IWES-Tandem, NREL 5MW, DTU 10MW RWT and LEANWIND.

### 2.1.2 Monopile design

For the foundation, only the distance between the mudline and the transition piece is known and one of the monopile designs for the DTU 10MW RWT described by Velarde and Bachynski (2017) is used as a reference. The wind turbine is located in





20 m deep water, and the reference design is selected accordingly. The monopile is scaled based on the overturning moment at
the mudline, which is calculated from the turbine thrust under rated conditions. This results in a uniform monopile with a length
of 65.4 m, a diameter of 7.11 m, and a wall thickness of 87 mm, giving a diameter-to-thickness ratio of 81.7, which is within
the limits specified in NORSOK-N-004 (2022). An apparent fixity model is used to account for soil structure interactions, as
described by Barltrop and Adams (1991). Assuming stiff clay as the seabed soil, the monopile is modelled with 0 DoF at a
distance of four times its diameter below the seabed. As the in seawater submerged length of the monopile influences its modal
behaviour, the mean seawater level (MSL) is calculated based on tide measurements at the nearest harbour.

### 2.1.3 Rotor

The scaling of airfoils and aerodynamic characteristics is performed using QBlade and Xfoil. Under a non-disclosure agree-
ment, the reference blade structure, airfoils, including NACA and FFA profiles, is used. The blade structure is replicated in
QBLADE v2.0.5 for the extraction of 360 polars and fine-tuning of the aerodynamic lift and drag coefficients to 10 million
Reynolds number, akin to the 10MW NREL offshore wind turbine. Post-creation of the blade, various aerodynamic coefficients
such as the 0-lift angle, angle of attack at f=0.7 below and above the angle of attack, normal force coefficients, drag coefficient,
and pitching moment coefficient are calculated in post-processing. The material properties of the blade are then linearly scaled
with the radius. The mass of the blade is scaled from the 10 MW RWT from NREL, with the assumption that the mass scales
linearly with the volume as the density of the wing remains constant. The scaling laws for volume and area, considering the
thickness ratio is equal to 1, led to the conclusion that the mass is proportional to the volume, resulting in the final scaling law
of mass being proportional to R × c², with R being the radius of the blade and c the respective chord length. Similarly, the
stiffness of the blade is determined with the cord scaling ratio to the power 4 for both the flapwise stiffness and the edgewise
stiffness.

### 2.1.4 Nacelle and hub

For the simulation model, the mass and inertia of the nacelle and hub are required. While the hub mass is known, the nacelle
mass for the target turbine is unavailable and is therefore assumed to match that of the LEANWIND turbine. To obtain the
corresponding inertia values for both components, values for the DTU 10M WRWT (Bak et al., 2013) turbine are scaled using
the radius of gyration, assuming geometric similarity between both turbines. Given the defined mass values, the scaled inertia
can be obtained from:

$$I = \frac{m}{\dot{G}_{ref}^2} \tag{1}$$

where $I$ is the desired inertia value, $m$ is the given components mass, and $\dot{G}_{ref}$ is the radius of gyration of the referenced
turbine.





### 2.1.5 Drivetrain

Conventionally, the drivetrain is modelled as a two-mass model within openFAST, simplifying the drivetrain as a torsional spring and damper between the generator and the hub. Thus, rotor inertia, drivetrain inertia, and torsional spring stiffness and damping are needed for the model. The rotor inertia is obtained from the multibody model of the scaled rotor. As the design specifications of the drivetrain are confidential, the generator inertia and the drivetrain torsional stiffness and damping are based on values from a medium-speed drivetrain of the DTU 10MW RWT, Bak et al. (2013). The generator inertia is scaled based on the assumption that the generator diameter scales linearly with the number of pole pairs and remains a constant mass density. This assumes an equal generator length independent of power and number of poles. The number of pole pairs is calculated from the rated rotor speed and gearbox transmission ratio, assuming the same nominal output frequency for the current. The drivetrain torsional stiffness is adjusted to obtain the same frequencies of the torsional drivetrain mode (free-free) using formulas described by Jonkman (2012). The gearbox efficiency is set to 95% in all operational regions.

### 2.1.6 Controller

The ROSCO controller, Abbas et al. (2021), is tuned using the OpenFAST toolbox, Branlard et al. (2023), and integrated into the model. The set-points of the pitch saturation routine are derived from available SCADA data to match the observed control behaviour. Further details about the controller cannot be disclosed as these are subject to confidentiality.

### 2.1.7 Mode shape coefficients

The OpenFAST sub-module ElastoDyn requires mode shape coefficients of the tower and blades to model the behaviour of the turbine. Mode shape coefficients can be obtained by measuring the displacement of several points along an excited structure. As no field measurement data with sufficient spatial resolution exists for the blades nor the tower, the mode shape coefficients are obtained with a multibody simulation approach. Within SIMPACK the tower and the blades are modelled and free decay tests are carried out, measuring the displacement of different points along the structure. The mode shape coefficients are then calculated using 6th-order polynomial fitting and the improved direct method described by Jonkman (2020). Mode shape coefficients for the 1st and 2nd tower bending modes, the 1st edgewise blade bending mode and the 1st and 2nd flapwise blade bending modes are obtained in this way.

### 2.2 System identification

The system identification aims to identify the low-frequency dynamic behaviour (i.e. eigenmodes) of the turbine from field measurements to tune and validate the modal behaviour of the model. For offshore wind turbines, environmental forces such as wind and waves excite the turbines's eigenmodes which can be measured as motion or other subsequent reactions, such as strain. For this cause, data from high frequency (4800 Hz) vibration measurements inside the nacelle of one turbine (T1) and low frequency (2 Hz) strain gauge measurements on the blades of three other turbines (T2-T4) of the same type are used. An OMA methodology is applied to these measurements to identify the first 13 main modes of the system, of which the first 10



are visualized in Figure 3 to 5. It is to note that the blade pitch angle was 90° during all measurements, resulting in the in-plane rotor modes (Figure 4) to include flapwise blade bending and the out-of-plane modes (Figure 5) to include edgewise blade bending, contrary to observations of an operating turbine.

In contrast to experimental modal analysis, the excitational forces (wind and wave) are not measured during the measurement of the strain and vibration data. To address this issue, a white noise excitation is commonly assumed within OMA, meaning the excitation forces are presumed to have an equally distributed power across a wide range of frequencies, Rainieri (2014). However, harmonic content from sources such as 1p and 3p excitations is present during the operation of WT, violating this assumption. Therefore only data from standstill periods of the turbine is used in the following approach. The applied OMA methodology can be divided into three steps:

1. Estimation of power spectral densities (PSDs) from the field measurements using Welch's method.

2. Automating the modal parameter estimation in the frequency domain.

3. Identifying modes and their corresponding eigenfrequencies.

Welch's method is used to obtain PSDs, as it provides a higher signal-to-noise ratio than a conventional Fourier transformation, by dividing the signal into overlapping segments, computing the Fourier transform of each segment, and then averaging the results. Increasing the number of averaged segments improves the signal-to-noise ratio by reducing the effect of random noise. However, since the frequency resolution of a Fourier transformation depends on the number of samples within each segment, increasing the number of segments decreases the frequency resolution of the PSD, given finite sample length. To achieve high noise reduction while preserving sufficient frequency resolution, a 90% overlap is chosen for segmenting.

Utlising the PSDs as an input, a modal parameter estimator in frequency domain is used to determine the eigenfrequencies of the system by identifying its stable poles. A least squares complex exponential (LSCE) method is used due to its robustness against noise, its computational efficiency, and the potential for automation. The estimator provides identified poles of increasing order in stability charts, categorizing them based on variation in frequency and damping. Poles are categorized as stable if the relative difference in frequency and damping values for poles of successive orders does not exceed certain thresholds, Brandt (2013). Clusters of stable poles indicate the presence of an underlying eigenmode of the system and allow to estimate its frequency and damping. This approach is automated in MATLAB to identify stable poles for all measurements collected within the standstill period. Since the measurement locations are concentrated within the nacelle or limited to a single section of the blades, the spatial resolution is insufficient to accurately describe the global mode shapes of the system solely based on the measurements. The underlying modes for the estimated eigenfrequencies are therefore identified based on their frequency, sensor orientation, domain knowledge and the utilisation of the multibody model. The method is first applied to the measurements of the drivetrain vibrations and then to the blade strain measurements to identify the eigenmodes of the turbine. A comprehensive description of this process is provided in the authors' previous work Gebel et al. (2024), which also details the utilized vibration measurement system in the nacelle.





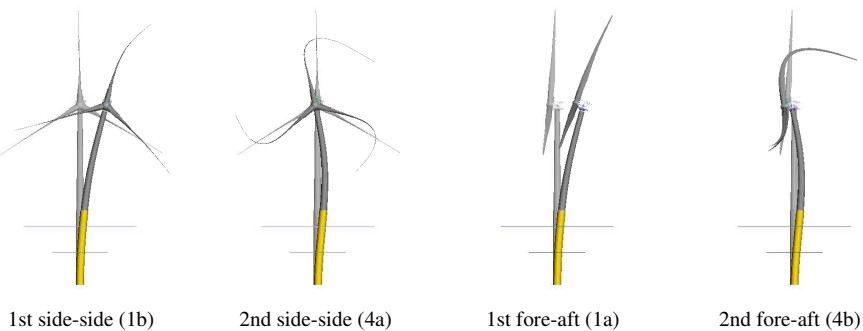

| 1st side-side (1b) | 2nd side-side (4a) | 1st fore-aft (1a) | 2nd fore-aft (4b) |

**Figure 3.** First and second tower bending modes for 90° blade pitch

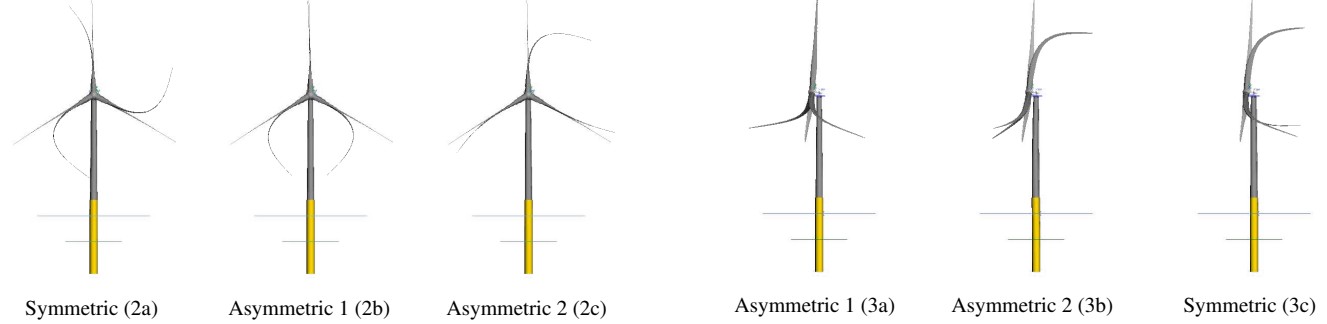

| Symmetric (2a) | Asymmetric 1 (2b) | Asymmetric 2 (2c) | Asymmetric 1 (3a) | Asymmetric 2 (3b) | Symmetric (3c) |

**Figure 4.** First in-plane (flapwise) rotor modes for 90° blade pitch

**Figure 5.** First out-of-plane (edgewise) rotor modes for 90° blade pitch

### 2.2.1 Modal parameter estimation from vibration measurements

To identify the system's modal behaviour, the vibration measurements of turbine T1 are used. During a standstill periods the vibrations inside the nacelle are measured with 16 sensors placed on different locations on the drivetrain. Sensors are placed on the main bearings, the gearbox housing and the generator housing in axial and radial direction of the main shaft. Ten minute time series are recorded at 4800 HZ every 15 minutes, resulting in 96 measurements during a 24 hour standstill period. The data is analysed with the described automated approach, defining a pole as stable if the relative difference in frequency of successive poles is smaller than 0.15 % and the relative difference in damping is smaller than 1.25 %. Poles are considered to belong to a physical mode, when at least three poles are identified as stable in consecutive model orders. The damping and frequency values of the identified stable poles of all measurement periods are plotted in Figure 6, divided into their origin from either axial or radial measurements. The PSDs are averaged over all measuremetns for both sensor sets. Each cluster of stable poles indicates one or more system modes to be present and allows to estimate its frequency and damping. Dashed vertical lines are located at the median value within each pole cluster and mark the frequency of possible eigenmodes with annotations





corresponding to Figures 3 to 5. The grey areas indicate the frequency ranges of 1P, 3P and 9P excitations, that would be present during operation of the turbine.

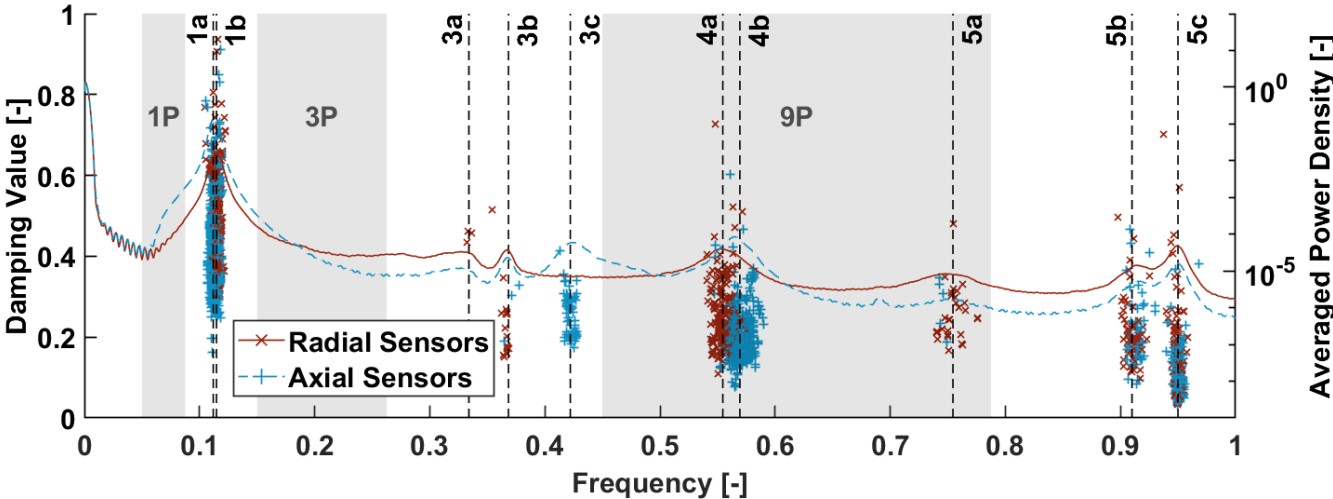

**Figure 6.** Normalized damping and frequency of stable poles and averaged PSDs of all measurements, split in radial and axial sensors.

For the identification of the modes, distinguishing the poles by the sensor orientation gives information about the modes
directionality and allows the separation of closely spaced modes. Modes 1a and 1b are identified as the 1st tower fore-aft (FA) and side-side (SS) bending modes, primarily visible in axial and radial measurements, respectively. Similary, are the modes 4a and 4b are identified as the 2nd tower SS and FA bending modes respectively. For the 2nd tower bending modes, a wider spread in frequency can be seen for the identified poles, this is related to their dependency on the tidal levels. As described in Gebel et al. (2024), the 2nd tower modes frequencies are lower during high tide, as the additional submerged length of the monopile
results in higher stiffness of the system due to added mass of the seawater. This effect is more pronounced for the 2nd bending modes as the location of maximum displacement is close to the water surface and not at the tower top, as the case for the 1st bending modes. The temporal changes of these eigenfrequencies are displayed in Figure 9 in Sect. 2.4.2. Modes at 3a and 3b are identified as asymmetric out-of-plane rotor modes 1 and 2, which induce nacelle pitch and yaw motions. To differentiate between the yaw-inducing mode 3a and the pitch-inducing mode 3b, the radial sensors are categorized into horizontally and
vertically oriented sensors. The symmetric out-of-plane rotor mode 3c primarily induces FA motion and is visible solely in axial measurements. Showing mainly in radial sensors, the mode at 5a is identified as the 2nd symmetric in-plane rotor mode while the modes at 5b and 5c are identified as the 2nd asymmetric in-plane rotor mode modes 1 and 2 respectively. The 1st in-plane rotor modes could not be found with this approach, as no notable peaks are found within the corresponding frequency range. Based on the reference turbines and the multibody model, the 1st in-plane rotor modes were expected to be located
between 0.15 and 0.3 of the normalized frequency.





### 2.2.2 Modal parameter estimation from blade strain data

To determine the frequencies of the 1st in-plane rotor modes, the blade stain measurements of the three turbines T2-T4 are analysed. The turbines are located in the same windfarm as T1 and are of the same type. The blade strain gauges are arranged in a ring around each blade, placed on the leading and trailing edge as well as on the windward and leeward sides of the blades.

The exact span-wise position along the blade is not known to the authors. The strain gauges were not calibrated at the time of the measurements, resulting in uncalibrated signal amplitudes. However, OMA does not aim to provide a scaled modal model, since the amplitude of the excitation is unknown, thus the frequency content of the measurements can still be used, as the scaling of the signal does not affect the estimated frequencies.

For the strain measurements, the analysis is separated into edgewise and flapwise measurements. The modal parameter

estimation identifies poles as stable if the frequency differs less than 0.25 % and the damping less than 1.5 % between poles of successive model orders. Figure 7 displays the frequency and damping values of identified stable poles for the edgewise and flapwise strain measurements and the PSDs for each turbine. In Figure 7a only poles at one frequency (2) are identified for the three 1st in-plane rotor modes (2a, 2b & 2c). As the amplitudes of the strain measurements are not calibrated, it is not possible to determine which of the three modes is the underlying mode for the identified poles. In Figure 7b poles are identified around

the frequency of the 1st out-of-plane rotor modes seen in the drivetrain vibration with a wide spread in frequency values and no reliable estimation can be reached from these measurements. The estimated frequencies for the first SS (1b) and FA (1a) tower modes align with the values obtained from the drivetrain vibration measurements. With the applied methodology, the measurement frequency of 2 Hz does not allow the identification of modal parameters above the Nyquist frequency of 1 Hz. The normalized frequencies of all identified modes from drivetrain vibration and blade strain data are summarized in Table 1.

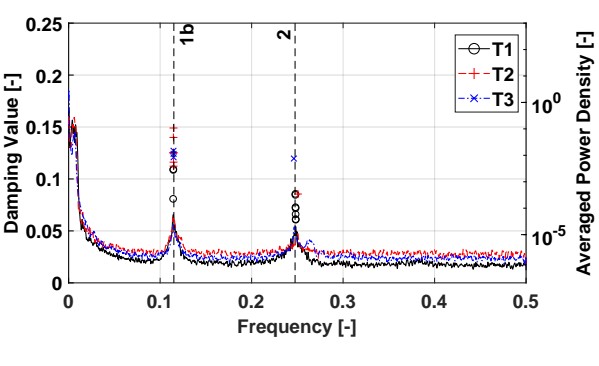
(a) Flapwise

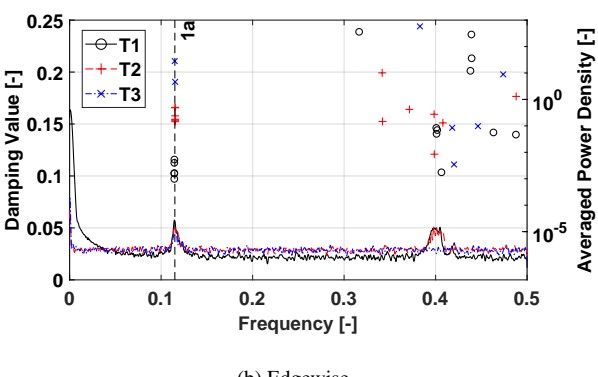
(b) Edgewise

**Figure 7.** Normalized frequency and damping values of identified stable poles from blade strain measurements with averaged PSDs of each turbine.



**Table 1.** Frequencies of identified modes from drivetrain vibration and blade strain data, and a comparison to the openFAST model in three different modeling stages, which are introduced in Sect. 2.3. Frequencies of not sufficiently separated modes are given in combined cells. Grey cells indicate frequencies of the models that differ more than 2 % from the identified eigenfrequencies.

| Mode | | | Drivetrain vibration [-] | Blade strain [-] | Initial basic model [-] | Tuned basic model [-] | Tuned extended model [-] |
|---|---|---|---|---|---|---|---|
| 1st tower bending | FA | (1a) | 0.112 | 0.114 | 0.104 | 0.113 | 0.112 |
| | SS | (1b) | 0.115 | 0.115 | 0.106 | 0.115 | 0.114 |
| 1st in-plane rotor (flapwise) | symmetric | (2a) | | | 0.221 | 0.231 | 0.229 |
| | asymmetric 1 | (2b) | - | 0.248 | 0.235 | 0.247 | 0.247 |
| | asymmetric 2 | (2c) | | | 0.238 | 0.251 | 0.250 |
| 1st out-of-plane rotor (edgewise) | asymmetric 1 | (3a) | 0.334 | | 0.410 | 0.407 | 0.332 |
| | asymmetric 2 | (3b) | 0.367 | - | 0.414 | 0.412 | 0.364 |
| | symmetric | (3c) | 0.423 | | 0.426 | 0.425 | 0.424 |
| 2nd tower bending | SS | (4a) | 0.558 | | 0.555 | 0.556 | 0.555 |
| | FA | (4b) | 0.569 | - | 0.576 | 0.578 | 0.578 |
| 2nd in-plane rotor (flapwise) | symmetric | (5a) | 0.754 | | 0.824 | 0.763 | 0.761 |
| | asymmetric 1 | (5b) | 0.915 | - | 1.089 | 0.940 | 0.916 |
| | asymmetric 2 | (5c) | 0.950 | | 1.093 | 0.945 | 0.933 |

## 2.3 Model tuning


This section explains the model tuning process and lists the adjustments made to the initial model. Table 1 allows to compare the eigenfrequencies of the initially scaled model from Sect. 2 to the obtained frequencies from the field measurements. Based on the found deviations, model properties, such as masses and stiffnesses are adjusted to tune the eigenfrequencies and the "tuned basic model" is obtained. An additional modelling step is applied to allow for main shaft bending and reduce remaining

deviations between the model and the deployed turbine, resulting in the "tuned extended model". The first tower bending modes (1a & 1b) of the initial model are 7 % to 8 % lower than the values obtained from the vibration data, while the 2nd tower modes (4a & 4b) are in good agreement. Therefore, instead of increasing the tower stiffness which affects the 2nd tower modes as well, the tower top mass is decreased by 17.5 %, affecting the 2nd tower bending modes only slightly.

As only one frequency value for the 1st in-plane rotor modes was found in the system identification, the obtained value is

used as a reference for all three modes to calculate the percental differences. Treating the blade as a linear beam the blade flap stiffness is adjusted using Equation 2, to increase the frequencies of the in-plane rotor modes.

$$f = \frac{1}{2\pi}\sqrt{\frac{k}{m}} \tag{2}$$





With $f$ being the eigenfrequency, $k$ the stiffness of the beam and $m$ the mass of the beam, the equation results in an estimated uniform increase of the flap stiffness by 11 % resulting in a good agreement of the asymmetric modes 1 and 2 (2b & 2c) while an error of 6.8 % persists for the symmetric mode (2a). It is seen in all other rotor modes that the symmetric mode is separated in frequency from the asymmetric modes. Therefore no additional tuning was applied to reduce the difference between the symmetric and the asymmetric modes here. It is to note that the frequency obtained from blade strain measurements could belong to the symmetric mode instead of the asymmetric modes as well. Using the 1st out-of-plane asymetric rotor modes 1 and 2 (5b & 5c) the stiffness of the blades is tuned for the 2nd flapwise modes. Their frequency is decreased by adjusting the blade flapwise modal stiffness tuner for the 2nd mode in ElastoDyn accordingly. As measurement values are available for the individual modes here, the symmetric out-of-plane mode (5a) can be tuned. This symmetric mode is used to tune the drivetrain torsional stiffness as it is the most sensible to changes in the drivetrain stiffness due to the collective motion of the baldes. The adjustment of drivetrain torsional stiffness influences the symmetric 1st in-plane rotor mode (2a) as well.

For the tuning of the edgewise bending stiffness of the blades, it is necessary to look at the symmetric out-of-plane rotor mode (3c). This mode does not induce mainshaft bending and can therefore be used to adjust the edgewise bending stiffness. As the eigenfrequencies of this mode are already in good agreement, no changes to the edgewise blade bending stiffnesses are necessary. The asymmetric out-of-plane modes 1 and 2 (3a & 3b) remain with significant deviations, this is attributed to the lack of mainshaft bending in the basic models, as these modes induce mainshaft bending due to their asymmetry.

To accurately model the differences between symmetric and asymmetric modes of the rotor, the model is extended with a flexible mainshaft, resulting in the "tuned extended model". The drivetrain, conventionally modelled as a two-mass system with a torsional spring and damper, is extended by flexible beam elements representing the main shaft by utilizing the openFAST submodel SuDyn. This modelling approach is described by Krathe et al. (2024) and requires the complete tower to be modelled in the SubDyn module instead of using ElastoDyn. The mainshaft dimensions are estimated based on the reference model of a 10MW-medium speed drivetrain and its bending stiffness is tuned based on the measured asymmetric out-of-plane bending modes (3a & 3b). The additional degrees of freedom allow to accurately represent the asymetric out-of-plane bending modes but also allow to keep the the error seen for both asymmetric in plane rotor modes (5b & 5c) below 2 %.

## 2.4 Model Validation

In the following sections, the performance of the obtained model is validated through simulations and compared to field measurements, including vibration and SCADA data. The turbine's operational behaviour is assessed against the provided power, pitch, and rpm curves from the operator. Additionally, the model's dynamic behaviour under varying tidal levels is compared to the observed behaviour from vibration sensors. These simulations aim to validate the turbine's operational performance and ensure consistency with real-world data.





### 2.4.1 Power Curve Comparison


As the power output of a wind turbine is one of its most important factors, the power output of the simulation is compared to the contractor power curve. The power output of the simulation model is calculated for wind speeds from cut-in to cut-out wind speed in 1 m/s steps using turbulent wind fields. A simulation length of 700 s is used, whereas the first 100 s are removed to avoid transient behaviour to influence the results, giving an effective simulation length of 10 minutes. To generate wind

fields for the simulations, turbulence intensities have to be known, these are obtained from six months of 1Hz SCADA data. Periods of curtailment or yaw misalignment are removed and the SCADA data is binned in 1 m/s bins. Average turbulence intensity values are obtained for each wind speed and wind fields are generated in turbSim, assuming normal IEC wind type and using the IECKAI turbulence model. A power-law shear profile with an exponent of 0.0721 is used. A surface roughness of $2 \times 10^{-4}$ m is chosen, corresponding to a calm open sea. As waves have a negligible influence on the power output of the

turbine, a still water surface is assumed. The resulting comparison is shown in Figure 8. The simulated values for pitch and rotor speed over wind speed are in similar agreement, but the respective curves cannot be shown due to confidentiality.

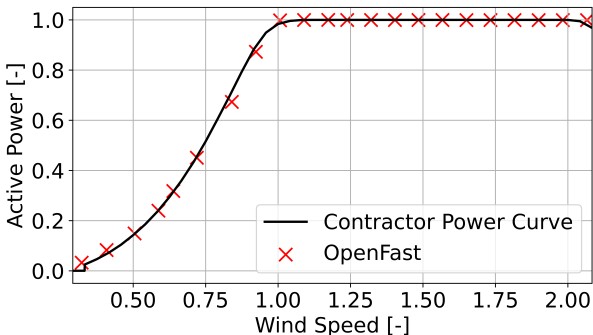

**Figure 8.** Power curve comparison between the power curve provided by the contractor and simulation results from OpenFAST using the described model.

### 2.4.2 Influence of MSL

As introduced in Sect. 2.2.1, the length of the monopile which is submerged in seawater influences its modal behaviour. The spectrograms in Figure 9 show the temporal changes of the 2nd tower bending modes frequency over 24 hours. To evaluate

whether the model's dynamic behaviour aligns with expected trends from the data under varying tidal levels, linearizations are conducted for different tidal conditions. Within the tuned extended model, the tidal level is set to the observed high tide level of 5m above LAT and the observed low tide level of 1m above LAT. The frequencies of the 2nd tower bending modes are calculated and compared to the frequencies obtained from the drivetrain vibration data. The results of the simulation model are indicated with red crosses in the the spectrograms in Figure 9. The simulation model's behaviour matches the deployed

turbine's overall behaviour while slightly overestimating the FA frequencies and underestimating the SS frequencies.





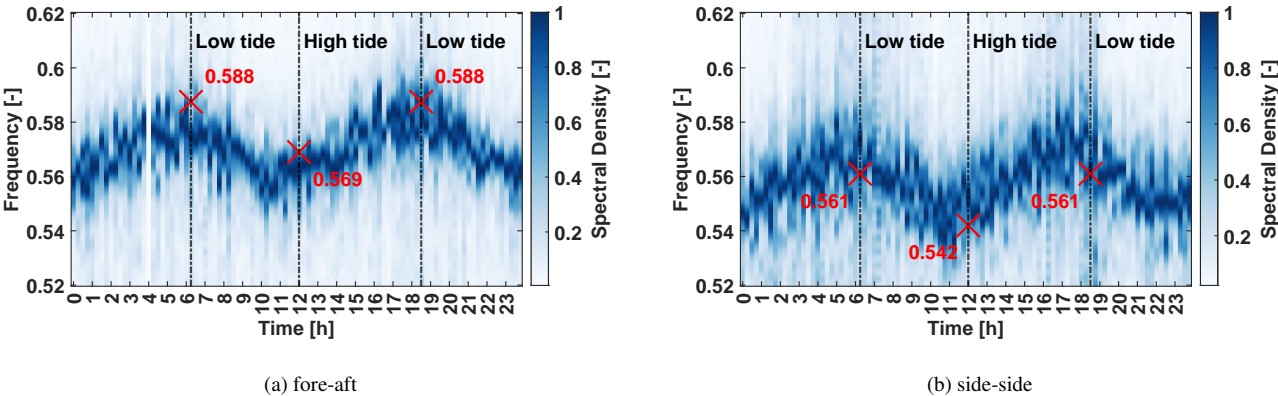

|                     |                      |
| :-----------------: | :------------------: |
| (a) fore-aft        | (b) side-side        |

**Figure 9.** Normalized spectrograms of the 2nd tower bending modes, as obtained from drivetrain vibration data. The simulation results for high and low tide are indicated as red crosses. Both spectral densities are averaged with respect to the highest value within each spectrogram.

## 3  Blade Pitch Bearing Lifetime Estimation

The following sections present a use case for the aero-servo-hydro-elastic model in calculating pitch bearing damage estimates under various environmental conditions. The simulation model provides load time series for the blade root during normal operation in different wind conditions. Namely turbulence intensities and wind shear of the inflow wind are varied, providing insights into the influence of these factors on pitch bearing deterioration. For turbulence intensity, the three standardized turbulence classes IEC-A, IEC-B, and IEC-C, as defined in IEC61400-1 (2019), are utilized. For wind shear, the exponent of the power law model is adjusted to create inflows with low, medium, and high wind shear, as specified in IEC61400-1 (2019) as well. Simulations are conducted for wind speeds ranging from cut-in to cut-out in 1 m/s increments. The resulting load time series are then used to calculate the rolling contact fatigue life of the pitch bearings using an analytical approach described in Sect. 3.1. The design specifications of the applied pitch bearing are found in Sect. 3.2 and results are presented in Sect. 3.3.

### 3.1  Pitch bearing rolling contact fatigue life calculation

Pitch bearings turn the blade around its longitudinal axis to change the blade's angle of attack. The pitch bearing moves approximately a quarter of a full rotation when going from blade feather towards production, and when in normal operation, only a partial amount of that motion is expected. The movement is a slow oscillation, which makes the periodicity that the classical calculation approach builds on disappear.

Pitch-bearing failures can be categorized into two main failure modes, surface-initiated and under-surface-initiated (Andreasen et al., 2022). Surface-initiated or wear modes consist of rotational wear, fretting, and false brinelling. Under-surface initiated or fatigue modes consist of ring fracture, edge loading, core crushing, and rolling contact fatigue. Rolling contact fatigue is one of the main failure modes, where all the possible damage starts at the contact point between the rolling elements



and raceways (Andreasen et al., 2022; Reinares et al., 2019). The rolling-contact fatigue of pitch bearings is the focus of the study.

Menck and Stammler (2024) reviewed different methods in rolling contact fatigue life calculation for oscillating bearings. They recommended the ISO-related rolling contact fatigue calculation. Among the ISO-related, the National Renewable Energy Laboratory's (NREL's) pitch and yaw bearing design guide (DG03) by Harris et al. (2009) is the most common guideline for

pitch bearing life calculation. DG03 is used in the wind industry and provides guidelines for determining the rating life of blade bearings (DNV, 2016). However, the new guideline Stammler et al. (2024) has recently been published. The new guideline undergoes several changes, and in this paper, whenever referred to DG03, the new guideline is intended.

The DG03 considers the pitch bearings as thrust-type bearings and uses the basic dynamic axial load rating, $C_a$, and the dynamic axial load, $\overline{P_{ea}}$, to estimate the basic rating life, $L_{10}$,

$$L_{10} = \left( \frac{C_a}{\overline{P_a}} \right)^p \tag{3}$$

where the life exponent, $p$, is equivalent to 3 for ball bearings. The basic dynamic axial load rating is defined as the constant centric thrust load that a rolling bearing theoretically could endure for a rating life, $L_{10}$, of 1 million revolutions.

$$C_a = 3.647 \times b_m f_c (i cos\alpha)^{0.7} Z^{2/3} D_w^{1/4} tan\alpha \tag{4}$$

where $Z$ is the number of balls, $i$ is the number of rows that carry an axial load simultaneously, $\alpha$ is a contact angle, and $D_w$

is the ball diameter. Variable $b_m = 1.3$ for axial ball bearings and $f_c$ is dependent on a groove conformity, pitch diameter, ball diameter, and contact angle, and it can be calculated from DG03. The dynamic equivalent axial load, $P_{ea}$, is formulated as

$$P_{ea} = 0.75 F_r + F_a + \frac{k_m M}{D_{pw}} \tag{5}$$

where $F_r$, $F_a$, and $M$ denote the applied radial, axial, and moment loads, respectively. $D_{pw}$ denotes the pitch diameter of the bearing. Due to the overturning moment in pitch bearings, the variable $k_M$ varies depending on the bearing. Menck et al.

(2020) adjust the term and the formula changes to

$$P_{ea} = 0.75 F_r + F_a + \frac{2.5 M}{D_{pw}} \tag{6}$$

Blade bearings experience a set of operations in different wind conditions. To include various duty cycle loading series of forces and moments were presented as bins. In each bin, the life is calculated, and the combined life of all bins, L, is then calculated as per

$$L = \frac{\sum_{k=1}^{K} q_k}{\sum_{k=1}^{K} \frac{q_k}{l_k}} \tag{7}$$

where $K$ is the number of bins and $l_k$ is the lives of each respective bin. $q_k$ is the timeshare on the k-th bin as

$$q_k = \frac{t_k \theta_k}{\sum_{i=1}^{K} t_i \theta_i} \tag{8}$$





Here, parameters $t_k$ denote the decimal fraction of the time of pitch movement for the bearing operated under the condition yielding $\theta_k$. The $L_10$ refers to the first visible damage on the raceways. The service life, $L_{10,srv}$, can exceed the rating life and is calculated by considering a corrective factor, $a_{srv}$ as

$$L_{10,srv} = \alpha_{srv} L_1 0 \tag{9}$$

The value of $\alpha_{srv}$ is between 2 and 3, and in this paper, it is considered 3.

The bearing life can be expressed as a fraction of the life, $l_k$, in different load cases as described previously and in Menck and Stammler (2024) and DNV (2016). Consequently, it is possible to define the damage level in each load case, $D_k$,

$$D_k = \frac{q_k}{l_k} \tag{10}$$

Details of the damage level are presented in the work by Nejad et al. (2014a).

Two different post-analyses are carried out in the paper: short-term and long-term analyses. The short-term analysis is referred to as a simulation with a short-term wind condition, i.e., a 10-minute simulation at a given wind speed. In this case, the wind speed distribution is not considered, and it is assumed that all wind speeds occur with equal probability. In long-term analysis, the site-specific wind speed distribution is considered for the loads and life calculation, giving a more realistic representation of the damage accumulation over the turbines lifetime. The effect of these two post analyses are in the parameter $t_k$ where in short-term analysis it is equal for all simulations while in long-term analysis, it calculated from wind speed distribution. The calculated life in hours is representing the bearing life with operational loads from the simulated conditions based on the wind speed charachtersitics.

### 3.2 Pitch bearing specification

The pitch bearing specifications are unavailable; therefore, an exemplary pitch bearing is designed. The bearing is designed according to the process introduced in the work by Rezaei et al. (2023). The designed-pitch bearing is a double-row, four-point contact ball bearing. It consists of an outer ring attached to the hub and an inner ring attached to the blade. The rings are separated by two raceways filled with balls that let the inner ring move with minimal friction. The pitch bearing main dimensions are stated in Table 2.





**Table 2.** Pitch-bearing specifications

| Parameter | Value | Description |
|-----------|-------|-------------|
| $D$ | 80 | Ball diameter [mm] |
| $\alpha$ | 45 | Initial contact angle [°] |
| $D_{pw}$ | 4790 | Bearing pitch diameter [mm] |
| $Z$ | 158 | Number of balls per row |
| $r_i$ | 42.4 | Inner raceway groove radius [mm] |
| $r_o$ | 42.4 | Outer raceway groove radius [mm] |
| $i$ | 2 | Number of rows |

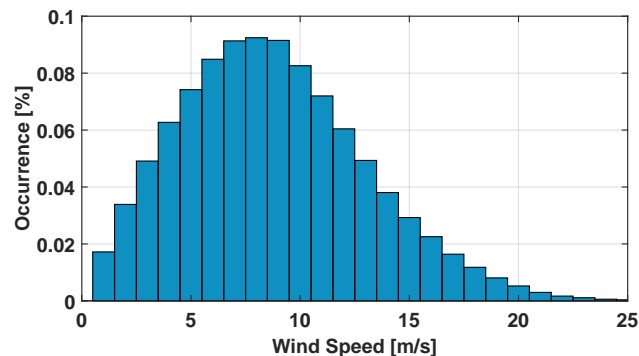

**Figure 10.** Histogram of wind speeds for 1980 to 2022 obtained from ERA5 data for the location of the deployed wind turbine

### 3.3 Influence of turbulence intensity and wind shear on pitch bearing damage

The different inflow wind conditions are shown in Figure 11, with the turbulence classes displayed in (a), as turbulence over wind speed and wind shear profiles in (b), as wind speed over heights with the corresponding power law exponents. The applied wind speed distribution for the long-term damages and total lifetime estimations is obtained from ERA5 data of the wind site 395 and given in Figure 10.

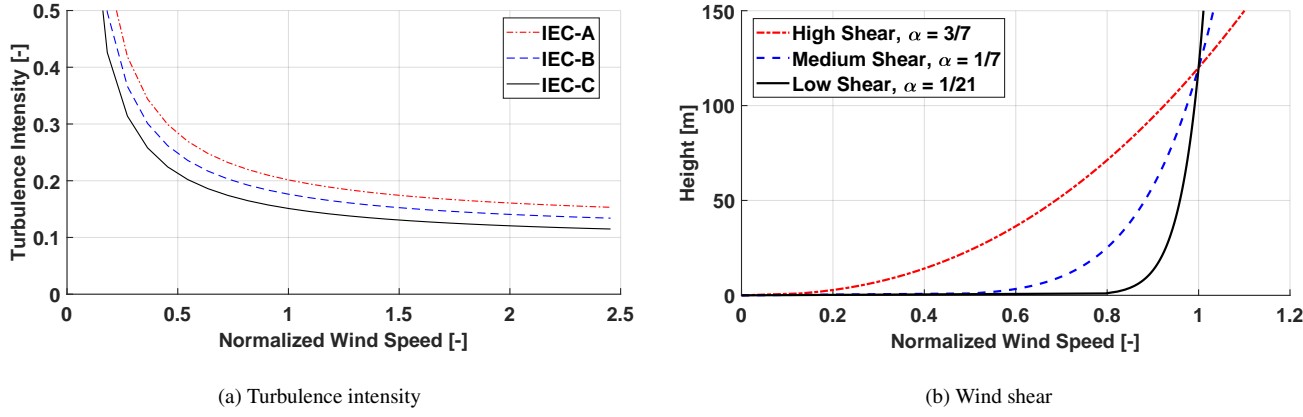

(a) Turbulence intensity                                     (b) Wind shear

**Figure 11.** Modifications to inflow wind conditions. Turbulence intensity values for the three applied turbulence classes in (a) and shear profiles with corresponding power law exponents in (b).





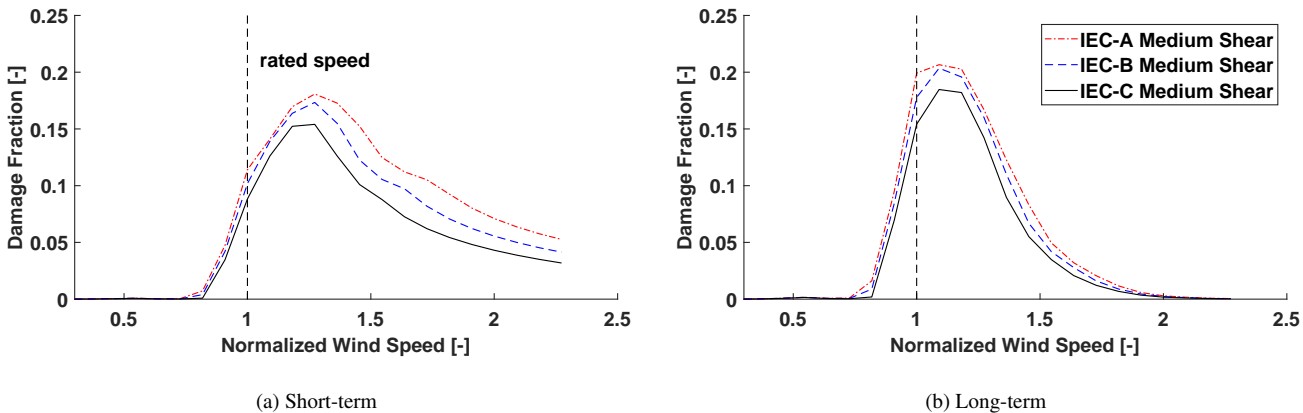

(a) Short-term

(b) Long-term

**Figure 12.** Normalized damage accumulation on blade pitch bearing for different turbulent intensities.

Figure 12, displays the estimated damage accumulation of the pitch bearings as damage fractions over wind speed for different turbulent intensities for short-term (a) and long-term (b) analysis. The wind speed is normalized with the rated wind speed of the turbine. The damage fraction indicates how much of the total lifetime is consumed in each windspeed bin. The integral of each curve can be interpreted as the accumulated damage after 20 years of normal operation. For an integral below

400   or equal to one, the demanded lifetime of 20 years is expected to be met, while a value above 1 indicates premature failure. The IEC-A turbulence model, which has the highest turbulence intensity, exhibits the highest damage fractions, while the IEC-C model consistently results in the lowest, indicating that higher turbulence intensity increases the pitch-bearing damage. This effect is more pronounced for above rated wind speeds. Although the highest thrust occurs at rated wind speed, the most damage per unit of time is seen slightly above rated wind speed for all three turbulence classes. This is attributed to the rotation

405   of the pitch bearings, on which the damage depends. Due to turbulence, the wind speed varies over time, when it surpasses the rated speed, the pitch angle is constant at 0 deg, thus reducing the number of pitch movements. This effect diminishes further away from the rated wind speed, resulting in a peak in the damage fraction slightly above the rated wind, where the thrust is still high but the rated wind speed is seldom surpassed. Considering the wind speed distribution of the site in the long-term damage, it is observed that wind speeds above twice the rated wind speed have a low contribution to overall damage. In addition

410   to relatively low damage fractions, their low occurrence results in a small contribution to the lifetime damage, while damage fractions around rated wind speed become more important due to their higher occurrence.

Given the wind speed distribution, the expected bearing lifetime results in 16.35 years for class IEC-A, 17.94 years for class IEC-B and 20.74 years for class IEC-C. It is to be noted that these estimations do not take dynamic start-up or shut-down events nor emergency breaks or standstills into account.



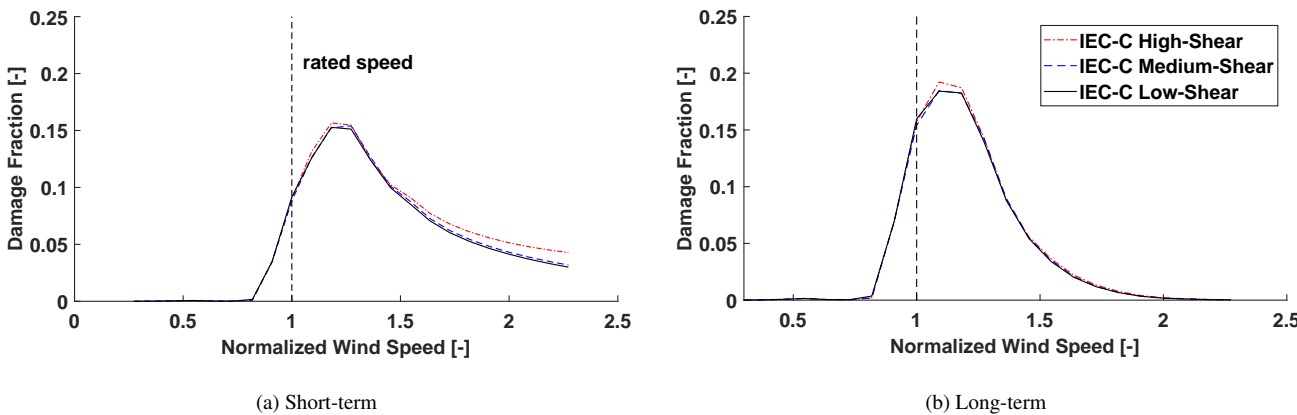

(a) Short-term  (b) Long-term

**Figure 13.** Normalized damage accumulation on blade pitch bearing for different wind shear.

Figure 13 shows the damage fractions of the pitch bearing for three different wind shear profiles in (a) the short-term and (b) the long-term analysis. Compared to turbulence intensity, the differences in damage due to wind shear are less pronounced. However, strong wind shear does result in higher damage fractions, particularly at wind speeds above 1.5 times the rated wind speed, where high-shear conditions produce noticeably more damage. This effect diminishes in the long-term analysis due to the low frequency of these high wind speeds. While the estimated lifetime differences for the turbulence cases are within a range of years, the differences for the wind shear cases are smaller. The estimated lifetime of 20.74 years for class IEC-C with medium shear is reduced by 168 days for high shear and prolonged by only one day for the low shear profile.

## 4 Conclusion

The work has shown how an aero-hydro-servo-elastic model of a deployed wind turbine can be obtained through model scaling and OMA. First, an initial model of a bottom-fixed offshore turbine has been derived by scaling reference turbines based on the available design parameters. Then, an OMA method in the frequency domain has been applied to drivetrain vibration and blade strain measurements to identify the system's eigenfrequencies. Here the first 13 eigenfrequencies of the system have been obtained and the respective modes identified. The eigenfrequencies have been used to calibrate the initial model, reducing the relative error between the model and the measured eigenfrequencies to less than 2 % for all but one mode. To achieve this, an additional modelling step has been introduced, extending the simplified drivetrain by a linear beam element, to allow for bending of the main shaft. Simulating normal operation, good agreement between the model and the SCADA data has been achieved regarding power, thrust and pitch behaviour. Additionally, the observed dependencies between tidal levels and the frequency variations of 2nd tower modes have been explained and replicated. The model now allows to simulate power production and structural loads in real-time, given imposed inflow conditions and can be used to understand the dependencies between inflow properties and component degradation.

In the second part of the paper, a first use case for the model has been presented, quantifying the detrimental effect of different turbulence intensities and wind shear profiles for pitch bearings. It has been shown that damage accumulation is the highest around rated wind speeds and negligible below 0.75 times rated wind speed in all cases due to its dependency on pitch-bearing movement. Examining the expected total component lifetime, minor differences have been observed for the three wind shear profiles, while significant differences have been observed for the three turbulence classes. This suggests pitch-
bearing damage to be more sensible to turbulence intensity than to differences in the shear profiles. While this holds true for the analysed conditions it is uncertain how the degradation behaves for other wind shear profiles and if the grade of variation in wind profiles is comparable with the variation in turbulence intensity.

    In future work, the model will be used to further examine the behaviour of fatigue damage of pitch bearings, blade roots and main bearings to provide a detailed overview of the degradational effect of different inflow properties and operational states.
The effects of operation with derated power output and the influence of partially waked turbines will be investigated to assess the costs of grid services and wake steering in form of turbine lifetime consumption. Possible applications of the model in a digital twin framework shall be explored with measured inflow properties based on LiDAR measurements to directly provide RUL estimates of turbine components as decision support for O&M.

## 5   Code availability

The code used in this study was developed in Python and MATLAB and cannot be made publicly available due to confidentiality.

## 6   Author contributions

JJM, JS, KV, and KK carried out the field measurements and preprocessed the data. Under the guidance of PJD, JG analyzed the data, developed the model code, and performed the simulations, while the pitch bearing fatigue, rotor scaling, and extended
mainshaft modeling were provided by AR, AV, and VLK, respectively. The manuscript was prepared by JG, with contributions from AR, AV, and VLK for their respective sections. PJD, ARN, and JH reviewed and edited the manuscript.

## 7   Competing interests

ARN is member of the editorial board of the Wind Energy Science journal. The authors declare that they have no further conflicts of interest.

## 8   Acknowledgements

The authors would like to acknowledge the financial support provided through the MaDurOS program by VLAIO (Flemish Agency for Innovation and Entrepreneurship) and SIM (Strategic Initiative Materials) as part of the SBO SeaFD project



(HBC.2019.0121). The authors would also like to acknowledge VLAIO for the funding of the CORE project. Furthermore, the authors acknowledge the Energy Transition Funds of the Belgian Federal Government for their funding of the BeFORECAST
and POSEIDON projects. Finally, the second and sixth authors are partially supported by Made4Wind project funded by the European Union under grant agreement 101136096.





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
