# Peer review of "System identification of offshore wind turbines for model updating and validation using field measurements"

_Wind Energy Science, 2024_

## Referee Comment (RC2)

The paper titled "*System identification of offshore wind turbines for model updating and validation using field measurements*" is well-performed multidisciplinary research focusing on two main concepts:

1) Tunning and validating generic offshore turbine model using condition monitoring system and SCADA

2) Estimating rolling contact fatigue of pitch bearings

Both above concepts are relevant to the current developments in wind industry and add high value by answering important questions specifically relevant for offshore lifetime extension where model accuracy becomes more important. The topic is becoming more crucial with first offshore wind farms reaching end of design life.

The work also includes several novel aspects including:

- Combine OMA with physics-based system knowledge to improve accuracy, including distinguishing closely spaced modes and filtering deterministic load and control effects when using this technique.

- Bridging between the measurements and analytical models for estimating lifetime of the bearing and study of the effect of environmental parameters on the damage consumption.

- Adding degrees of freedom in modelling the shaft to reduce the errors in estimating the modal parameters

The work is worth publishing in the Journal of Wind Energy Science after few modifications listed below.

**Major comments:**

1) The manuscript would benefit from clearer boundaries between the two main topics (model development and bearing lifetime assessment), each of which could stand alone as independent research. Please make this distinction explicit—particularly in the abstract and introduction.

2) In the second part of the research, the wind speed distribution of the site is used while keeping TI of the class for defining the Rolling contact fatigue (RCF) lifetime of the pitch bearing. The rationale for this choice is unclear. If the aim is a site-specific assessment, why not use the site's measured turbulence? If the aim is solely seeing the effect of inputs in each class, why not fully commit to class input levels? Please make the justification explicit in the manuscript.

3) The study of the effects of environmental parameters on the fatigue life of the bearing needs stronger base and elaborations for the reader. While the results are valuable, they do not seem to be continuation of the flow of the study (site-specific versus design for the case study turbine using the tunned and validated model would have been more connected). Please consider making a flow through explanations in the text or justifications based on the response to comment #2 above.

4) Suggestion: As the full measurement of environmental parameters are in hand, for the OMA purpose, the responses in standstill state can be filtered to include mostly durations with low turbulence. This way the condition could have been closer to the white noise assumption in OMA. Please either apply or comment.

5) The validations should have been based on load response and/or other model outputs instead of just power curve. Although other parameters (pitch and rpm) are mentioned as considered, only presenting the power curve does not add high value. Please consider validating against a parameter which is not obtained already in the scaling step. If the concern is maintaining confidentiality, normalizing and/or tabular presentation of percentage differences can be helpful. Please also consider that investigating the scatter of output (variance) with scatter of input (e.g. high and low tail of turbulence in each MWS) is important (not only looking at the mean output and mean TI as stated in the paper).

   *Suggestion: It is understood that the strain gauges are not calibrated but a track of fatigue load responses via measurements under two specific operational conditions (in which the controller is active) followed by doing the same using the aeroelastic tunned model can show whether the model shows a similar ratio/regression in the response. This shows the validity of the model to be used for fatigue load comparisons under different environmental inputs (which is common is assessing lifetime extension).*

   If you skip doing the suggestion, you should mention the load validation as a missing point and suggestion for future work.

6) This type of work (especially tuning of the generic models) includes a lot of uncertainty based on different assumptions and the missing information. A discussion section maneuvering on different aspects (e.g. scaling method, assumptions, methods used and measurements) are their possible effects that are needed and will add value. This is especially important if presenting other variables for validation (see major comment #5) remain missing.

   Suggestion: The study 'Robertson, Amy N., et al. '*Sensitivity analysis of the effect of wind characteristics and turbine properties on wind turbine loads*', Wind Energy Science 4.3 (2019): 479-513.' can be a good base for finding areas to focus on in the discussion regarding model bias or uncertainties and their effect in the current work. The applicability of the model for site-specific fatigue assessment is a better match for consideration in the current paper (due to lifetime extension and fatigue assessment of the pitch bearing).

7) The effects of turbulence can be considered predictive as more turbulence often brings more control (pitch) activities and thus more contact rolling fatigue, no? Please consider the discussions.

**Minor comments:**

1) Overall, the paper is wordy, particularly in the introduction, which has room for more briefness. Throughout the manuscript, it is difficult to keep track of both the big picture and the detailed points at the same time. To improve readability and flow, I suggest restructuring: consider

placing all high-level descriptions in one section (e.g., Methodology) and then presenting the detailed assessments in the subsequent Results section. This may also address the concern raised in Major Comment #1 as it can provide more focus in introduction on the flow of the work.

2) The reason for the 1st in-plane blade mode not being detected via high frequency measurements while out-of-plane is detected is worth discussing in the paper. Is it the orientation of the sensors? Is there any correlation between having mostly turbulence related excitation in T1 turbine location during stand-still? Or any other reasons? Please consider explaining in the text.

3) Please list all the design parameters available at the beginning of section 2. This information is highly important while floating in different places in the text. Consider listing or even making a table introducing them.

4) Section 2.1.3: Is the actual profile of the turbine in hand? This needs a bit more clarity in wording to understand

5) Omitting the word edgewise and flapwise and sticking to in- and out- of plane will improve the readability and clarity. This can be clarified once for the whole paper (when describing the 90 degrees pitch).

6) Please elaborate on the measurement tools used for environmental data collection as well as their placement (in relation to each of the four turbines).

7) Line 439: the effect of mean wind speed is worth mentioning.

**Technical corrections:**

1) Please correct the typos below:

   Line 92: *"eingenfrequencies"* → "eigenfrequencies"
   Line 221: *"Similary"* → "Similarly"
   Line 278: *"baldes"* → "blades"
   In few places in the document (please search and correct all): *"asymetric"* → "asymmetric"
   Line 288: *"suDyn"* → "SubDyn"
   Line 191: *"Utlising"* → "Utilising"
   Line 384: *"charachtersitics"* → "characteristics"

2) Please delete the extra **'the'** in line 292

3) Line 231: please reword below sentence for more clarity and briefness:

'*Showing mainly in radial sensors, the mode at 5a is identified as the 2nd symmetric in-plane rotor mode while the modes at 5b and 5c are identified as the 2nd asymmetric in-plane rotor mode modes 1 and 2 respectively.*'

4) Line 213: The phrase '*at least three poles are identified as stable in consecutive model orders*' is not clear.

---

## Author Comment (AC1)

**Review No 1**

**Major comments:**

1) The article is the combination of two fairly distinct investigations. In particular, the tuning of the model described in the first part is likely quite irrelevant to the second part where the lifetime of the bearings is estimated. In the first part of the article you tune the elastic properties of blades and tower. These do not seem to be the key drivers of loads in the pitch bearings. Is it a good idea to combine the two stories together? Please discuss.

Following the feedback of the reviews, the paper will be shortened to maintain sole focus on the tuning and model description. The damage calculation is covered in another paper.

2) Related to point #1, the power curve comparison presented in section 2.4.1 is not really a validation. If you showed the initial basic model in Figure 8, I believe it would overlap with the red markers generated by the tuned model. The power curve is only mildly influenced by the natural frequencies of the system (except if major instabilities are present, but that's probably not your case). So I don't think this section should be part of section 2.4 Validation. The only real validation is presented in Section 2.4.2.

The power curves will no longer be part of the validation. The validation will instead be extended by comparison of simulation results to field data during operational conditions. Specifically the root mean square of vibration signals over the operational regime of the turbine are compared to the simulated vibration at the tower top.

3) The paper only has 4 sections. I think readability would improve significantly if you could split the narrative into more sections. One idea could be to do: Section 2 Model Generation Section 3 System Identification Section 4 Model Tuning Section 5 Model Validation. If you do so, remember to update the text at line 78.

The sectioning will be revised following this suggestion.

4) The whole scaling process is pretty crude, but, more importantly, it is not well documented. At line 107, what does it mean that "tower bending stiffness is scaled based on rated thrust"? Same for line 113: what does it mean "scaled accordingly"? Section 2.1.3 is also nebulous. I understand that data is confidential, but why do you use such a complex tool like QBLade to do a Viterna extension of the polars? What does it mean that "material properties are linearly scaled"? Or "thickness ratio is equal to 1"? It seems that you simply scaled blade mass by the cube of the length. Am I missing anything? Overall, I would recommend a substantial rewriting of these sections, adding some rigor to the description of the scaling process.

The scaling processes will be described in clearer detail, providing utilized formulars and rigor information.

The section about the monopile will be revised completely, as the design data of the monopile was made accessible since the submission of the paper, such that a scaling of the monopile is no longer necessary.

For the blades (section 2.1.3) design details, such as the blade dimensions, cross sections and weight, are available to the authors. The availability of this information is already confidential. The authors would like to keep it that way, if possible. In any case, the section itself will be revised to provide a more clear procedure of the scaling approach and how one could reproduce this process.

5) Table 1: at 0 rpm you should not have rotor whirling modes. Why do you have 3 distinct natural frequencies for flap and 3 for edge? Also, I don't think you discuss how you've estimated these numbers. The comment also applies to line 426: I don't think you've obtained the first 13 eigenfrequencies. I think it's 7.

The found eigenfrequencies are indeed not from whirling rotor modes. The nomenclature of the modes might be misleading and thus the observed modes will be renamed to follow the nomenclature here:

Figure 3.5: The 1st rotor modes of a wind turbine: side view for the flapwise modes and front view for the edgewise modes (undeformed and deformed models are shown).

**Found at page 62 of:**

Peeters, Joris. "Simulation of dynamic drive train loads in a wind turbine." Katholieke Universiteit Leuven (2006).

**Minor comments:**

- The introduction is too verbose. You submitted your manuscript to a wind energy journal, there is no need to talk about global deployment goals. Also because these get very quickly outdated: the USA have just pivoted away from offshore wind (your citation at line 16 is outdated). Please rethink the text between lines 13 and 40 and make it more specific to your manuscript. Please focus on your contributions that are of interest to your readers.

The introduction will be reworked following this comment to make the introduction more terse and relevant to the presented work.

- Line 68: there is a major difference between damage equivalent loads and lifetime estimates, which are usually based on stress metrics. Aero-servo-hydro-elastic models estimate DELs, but for lifetime more is needed. Please highlight this critical difference.

Will The difference between DELs and lifetime estimations will be adressed

- Line 85: "around" 8MW? I imagine rated power is not proprietary...

Will be changed to more specific 8.4MW

- Line 86: This is maybe not of critical importance, but it's surprising to see an expensive commercial tool such as Simpack being used solely for visualization and modal characterization... Although in a clunkier fashion, but NREL tools do both things.

The work indeed could have been carried out mainly in openFAST, but the user interface of Simpack made the process more comprehensible. Especially the calculation of the mode shape coefficients and the visualization of the modes, as tools like ACDC for openFAST were not as developed as they are now, when the work was carried out.

- Line 104: the LEANWIND model could not be validated, when validation means compared to real-world data, since the LEANWIND model was purely theoretical. Note that the citation Desmond et al. 2013 is misleading, because it points to a DNV deliverable that I cannot find online (is it even publicly available?). I would recommend rephrasing this paragraph.

The paragraph will be rephrased and an accessible reference will be provided.

**- Line 116: what does "fixity" mean?**

The term "apparent fixity" describes a modelling method for the soil-structure interactions. It will be made clear that the apparent fixity model describes the monopile to have 0 degrees of freedom at a given depth below the sea bed, as it currently reads as two separate statements.

- Line 117: define DOF. Also, I think you can better explain that 0 degrees of freedom mean rigid clamping.

The term rigid clamping will be added as well as a definition of the DoF.

- Line 170: it's not entirely clear why you used 4 turbines and not just one. What is the reasoning and value of using 4? For example, the results do not clear characterize turbine to turbine variability.

Blade strain measurements are available for three turbines, whereas none of them have vibration measurements. To minimize the influence of turbine-to-turbine variability, the average modal frequencies derived from the strain measurements are used. It will be clarified in the paper that the applied methodology does not allow an exact match with the frequencies of the individual deployed turbines. However, it enables the development of a tuned model whose modal frequencies fall within the range of variability.

- Line 175: I find confusing that you interchange out-of-plane with edgewise and inplane with flapwise. I understand that the blades are parked, but I would stick to the words "edgewise" and "flapwise".

The modes will be called flapwise and edgewise consistently now, giving a note at the beginning that the blades are at 90deg, such that, contrary to an operating turbine, out-of-plane bending modes are not flapwise, but edgewise modes.

- Line 191: typo "Utilizing"
- Line 207: what kind of sensors are you using? did you install them, or did they come installed from the manufacturer? These are important details for replicability. For example, can your approach be replicated on any turbine, or does it require the installation of specific instrumentation?

For the vibration data, we use an IPC accelerometers which were installed by us. For the blade strain measurements we obtained data from strain gauges installed by the operator. More information of the utilized sensors and their installation will be provided

concerning the type of sensor as well as their exact location on the drivetrain and the blades.

- Line 209: typo "Hz"
- Line 214: typo "measurements"
- Figure 6: I don't see where you've defined how you've normalized Frequency

The frequency is normalized with a random value to keep confidentiality promises. This normalization will be pointed out in the revised paper to avoid misunderstandings. Or the normalization can be based on the frequency of the highest modes instead, if this is preferred.

- Line 229: I think I understand what a "yaw-inducing" mode is, but I am less sure about the "pitch-inducing" mode? Maybe better to link it back to Figure 5?

While this part will be removed, the yaw, and pitch, inducing modes will be explained a bit more to avoid misunderstanding.

- Line 236: typo "strain"
- Line 328: "as well"? Where else are these parameters specified?

This part will be removed

- Line 334: "The movement is a slow oscillation, which makes the periodicity that the classical calculation approach builds on disappear." What does this mean?

This part will be removed

- Line 405: "Due to turbulence, the wind speed varies over time, when it surpasses the rated speed, the pitch angle is constant at 0 deg, thus reducing the number of pitch movements." What does this mean? Same for "where the thrust is still high but the rated wind speed is seldom surpassed."

This part will be removed

---

## Author Comment (AC2)

**Review No 2**

**Major Comments**

1) The manuscript would benefit from clearer boundaries between the two main topics (model development and bearing lifetime assessment). Each could stand alone as independent research. Please make this distinction explicit in the abstract and introduction.

Following the feedback of the reviews, the paper will be shortened to maintain sole focus on the tuning and model description. The damage calculation is covered in another paper.

- 2) In the second part, the wind speed distribution of the site is used while keeping TI of the IEC class to define the rolling contact fatigue (RCF) lifetime of the pitch bearing. The rationale for this choice is unclear.
- If the aim is site-specific assessment, why not use measured turbulence?
- If the aim is to explore input classes, why not fully commit to class input levels? Please justify explicitly.

The chapter on fatigue life calculations will be removed from the paper

3) The study of environmental effects on bearing fatigue life requires a stronger basis and elaboration. While the results are valuable, they do not flow naturally from the earlier sections. Please improve the narrative link (site-specific vs design assessment) or justify clearly based on the response to Comment 2.

The chapter on fatigue life calculations will be removed from the paper

4) Since full environmental measurements are available, responses in standstill states could be filtered to include mostly low-turbulence periods. This would better approximate the whitenoise assumption in OMA. Please either apply or comment.

The standstill period was explicitly chosen for a duration of high wind speeds, during low wind speeds the excitation of the system is very small, which results in difficulties to separate modes from background noise. The 24-hour period which was used now, is the only long period with varying but high wind speeds, for which the turbine was in standstill.

- 5) Validation should not rely solely on the power curve. Although pitch and rpm are mentioned, only presenting the power curve adds little value. Please consider:
- Validating against other outputs not directly used in scaling.
- Normalizing/tabulating percentage differences if confidentiality is a concern.
- Investigating scatter of outputs (variance) vs scatter of inputs (e.g., turbulence extremes).
- Validating against other outputs not directly used in scaling:
- The power curves will no longer be part of the validation. The validation will instead be extended by comparison to field data of vibration signals during operational conditions. Specifically the root mean square of the vibration signals over the range of operational wind

speeds will be compared between the model and field measurements.

- Normalizing/tabulating percentage differences if confidentiality is a concern.
- It will be discussed with the operator if we can publish pitch and rpm curves as normalized curves.
- Investigating scatter of outputs (variance) vs scatter of inputs (e.g., turbulence extremes).
- Is this referring to the study on bearing lifetime assessment or model development? In either case, this can be provided as a series of simulations at the end of the respective paper.

Suggestion: Even with uncalibrated strain gauges, tracking fatigue load responses under two operational conditions and comparing with the tuned model can demonstrate validity for fatigue load comparisons. If not implemented, please mention this as a limitation and suggestion for future work.

It will be explored if it is feasible to compare load responses or frequency spectra of the strain gauges during operation with simulation results to obtain fatigue load comparison. This is most probably a topic for future work.

6) Tuning generic models involves uncertainty due to assumptions and missing information. A dedicated discussion is needed on scaling methods, assumptions, measurement limitations, and their effects.

Suggestion: Robertson et al. (2019), \*Sensitivity analysis of the effect of wind characteristics and turbine properties on wind turbine loads\* (\*Wind Energy Science\*, 4:479-513), could serve as a base to highlight potential biases. This would strengthen the applicability for site-specific fatigue assessments.

Thank you for already suggesting literature. A discussion on this will be added in the revised paper, as there is room for more targeted discussion about model tuning after the bearing fatigue was removed.

7) Please expand the discussion: more turbulence often induces more pitch activity, increasing RCF. Explicitly address this link.

The chapter on fatigue life calculations will be removed from the paper, but the feedback will be taken in mind for the work about bearing fatigue.

**Minor Comments**

1) The manuscript is overly wordy, especially in the introduction. Restructuring into a clear Methodology → Results flow would improve readability and may also help with Comment 1.

The introduction will be revised, being more terse and clear on the aim of the paper as well as its structure.

2) The absence of the 1st in-plane blade mode in high-frequency measurements (while out-of-plane is detected) should be explained. Could it relate to sensor orientation, turbulence characteristics, or other reasons?

The link between sensor placement and visibility of this mode will be discussed. The 1st inplane mode induces drivetrain torsion, which is does not result in a distinct frequency peak in the sensors, due to their orientation. No high frequency rpm measurements are available.

3) List all design parameters at the beginning of Section 2, ideally in a table.

A table will be added for a clear overview.

**4) Section 2.1.3: Please clarify whether the actual turbine profile is available.**

For the blades (section 2.1.3) design details, such as the blade dimensions, cross sections and weight, are available to the authors. The availability of this information, however, is already confidential itself. We see that this availability should be disclosed for the paper to be reproducable and will discuss with the corresponding parties.

5) Avoid "edgewise/flapwise"; use "in-plane/out-of-plane" consistently.

To follow the nomenclature advised in:

Peeters, Joris. "Simulation of dynamic drive train loads in a wind turbine." Katholieke Universiteit Leuven (2006).

The paper will use edgewise and flapwise exclusively and clarify that the blades are at a 90 deg pitch angle at all times at the beginning of the section, to avoid misunderstanding.

6) Provide more detail on environmental measurement tools and placement relative to the turbines.

For the vibration data, IPC accelerometers are used which were installed by us. For the blade strain measurements we obtained data from strain gauges installed by the operator. More information of the utilized sensors and their installation will be provided concerning the type of sensor as well as their location on the drivetrain and the blades.

**7) Line 439: The effect of mean wind speed should be mentioned.**

The chapter on fatigue life calculations will be removed from the paper, but the feedback will be taken in mind for the work about bearing fatigue.

**Technical Corrections**

- 1) Line 92: "eingenfrequencies" → "eigenfrequencies"
- 2) Line 191: "Utlising" → "Utilising"
- 3) Line 221: "Similary" → "Similarly"
- 4) Line 278: "baldes" → "blades"
- 5) Line 288: "suDyn" → "SubDyn"
- 6) Line 384: "charachtersitics" → "characteristics"
- 7) In several places: "asymetric" → "asymmetric" (please search/replace throughout).
- 8) Line 292: delete extra 'the'.
- 9) Line 231: Please reword for clarity: 'The mode at 5a is identified as the 2nd symmetric inplane rotor mode, while 5b and 5c are identified as the 2nd asymmetric in-plane rotor modes 1 and 2, respectively.'
- 10) Line 213: Clarify the phrase 'at least three poles are identified as stable in consecutive model orders.'